# Awareness and knowledge associated to Human papillomavirus infection among university students in Morocco: A cross-sectional study

**Nezha El Mansouri**[1]*, **Laila Ferrera**[1]°, **Ahmed Kharbach**[2]°, **Abderrahmane Achbani**[1‡], **Farid Kassidi**[1‡], **Hanane Rogua**[1‡], **Sofiane Ait Wahmane**[1‡], **Ahmed Belmouden**[1‡], **Said Chouham**[1‡], **Mohamed Nejmeddine**[1]*

**1** Laboratory of Cell Biology and Molecular Genetics (LBCGM), Department of Biology, Faculty of Sciences, Ibn Zohr University, Agadir, Souss Massa, Morocco, **2** Laboratory of Biostatistics, Clinical Research and Epidemiology (LBRCE), Faculty of Medicine and Pharmacy of Rabat, Mohammed V University, Rabat, Rabat-Salé-Kénitra, Morocco

° These authors contributed equally to this work.
‡ AA, FK, HR, SAW, AB and SC also contributed equally to this work.
* nezha.elmansouri@edu.uiz.ac.ma (NEM); m.nejmeddine@uiz.ac.ma (MN)

**Data Availability Statement:** All relevant data are within the paper and its Supporting information files.

## Abstract

Worldwide, cervical cancer is a real health issue, however, gaps exist in the public's awareness of the causal role of Human papillomavirus (HPV) in the development of this disease. This study aims to determine the level of awareness, knowledge and the associated factors on HPV among university students in Morocco. A cross-sectional study was conducted with a descriptive and analytical aim, among students attending Ibn Zohr University, in Agadir, Morocco. An interview questionnaire was used to collect information about the participants: demographic data, awareness and level of knowledge on HPV infection, and awareness of cervical cancer. Logistic regression analyses were used to determine the associated factors with awareness and level of knowledge on HPV. A total of 479 students participated in this study (mean age 21.82 ± 2.091). Most participants n = 391 (81.6%) were aware of cervical cancer, while only n = 7 (1.5%) identified HPV as a sexually transmitted infection. Among students, 10.0% (n = 48) were aware of HPV but only half of them n = 23 (47.9%) confirmed that HPV is associated with cervical cancer, and n = 29 (60.4%) showed low knowledge on HPV. Multivariate analysis revealed that HPV awareness has a strong association with a higher level of education (OR 4.04; 95% CI: 1.92–8.52), and with being a biology student (OR 5.20; 95% CI: 2.12–12.73), while high HPV knowledge was only associated with the female gender (OR 3.76; 95% CI: 1.01–13.92). The data suggest that university students in Morocco did not show sufficient knowledge of HPV infection and its consequences. This supports that earlier incorporation of sexual health education programs, especially related to HPV and cervical cancer, must be implemented in the university to reduce the burden of HPV-associated diseases among the population at risk.

**Funding:** The authors received no specific funding for this work.

**Competing interests:** The authors have declared that no competing interests exist.

## Introduction

The papillomavirus belongs to the family of papillomaviridae [1]. They co-evolved over millions of years with a diverse range of animal hosts as well as humans [2]. Human papillomavirus (HPV) is one of the most common causes of sexually transmitted infections (STIs), with the highest rates are found among young and sexually active men and women worldwide [3, 4]. HPVs constitute a group of more than 200 different types associated with benign and malignant neoplasms of the skin and mucosal membranes. Among them, forty different HPV types, known to infect the genital system. These HPV types are subdivided into low-risk types, which cause mainly genital warts; and high-risk types, which were reported to be responsible for almost all cases of cervical cancer (>99.7%) [5]. HPV 16 and HPV 18 are the most carcinogenic HPV types, causing approximately 70% of all cervical cancer cases. Three vaccines with a good safety profile were developed and were shown to be an effective primary prevention strategy against the most common HPV strains. Bivalent (HPV-16 and HPV-18), and quadrivalent (HPV16, 18, 6, and 11), and nonavalent vaccine(HPV 16, 18, 6, 11, 31, 33, 45, 52, and 58) [6]. HPVs were also shown to be associated with 30–60% of head-and-neck cancers, and cause different types of anogenital cancers [7]. Most HPV infections of the cervix are asymptomatic and are cleared within two years [8]. However, cervical cancer is mainly a consequence of the persistence of carcinogenic infections of HPV [9].

Cervical cancer is the fourth most frequently diagnosed cancer in women worldwide, with an estimated 604,000 new cases in 2020, representing 6.5% of all female cancers. It is the fourth leading cause of cancer death in women, with 342,000 deaths worldwide, according to the latest report from the International Agency for Research on Cancer (IARC) in 2020 [10]. This cancer occurs mostly in low and middle-income countries. The highest incidence rates have been recorded in Eastern Africa, while incidence rates are 7 to 10 times lower in North America, Australia/New Zealand, and Western Asia (40.1 per 100,000 in Eastern Africa vs 5.6 per 100,00 in Australia/New Zealand) in 2020 [10].

Cervical cancer is a major health issue in Morocco, with 2165 new cases were diagnosed in 2020. It also comes in the second position, after breast cancer is the leading cause of death by cancer in women, with 1199 death in 2020 [10].

The high frequency of cervical cancer, representing 7.2% of all the female cancers in the country, might be due either to the limited access to health care coverage especially in remote areas in the southern regions of the country or to poor awareness and knowledge about this illness as barriers to use preventive methods [11]. Despite these statistics, cervical cancer is one of the few preventable, and curable human cancer if detected earlier. It could be prevented by primary prevention (large vaccination programs), and secondary prevention approaches (early diagnosis by screening, and by implementing effective treatment programs). In addition, promising prevention strategies based on educational programs to improve knowledge of the pathogenesis of HPV and cervical cancer [12].

In Morocco, there is a lack of data on awareness and knowledge among the general public, on what causes cervical cancer. In this present study, we aim to investigate the level and the associated factors of awareness and knowledge on HPV among higher education students, attending Ibn Zohr University in Morocco, as young adults at risk of contracting HPV infection.

## Materials and methods

### The study setting and population sample

This cross-sectional study was conducted on the site of the faculty of sciences, Ibn Zohr University (IZU), in Agadir, Morocco, where the majority of the students were attending their curricula. A total of n = 479 participants were selected using a simple random selection process.

The sample size was calculated based on a 5.0% error range, a 95% confidence interval (CI) for a total population of 7000 students in the faculty of sciences, university Ibn Zohr. With an anticipated proportion of knowledge and awareness on HPV deficiency of 50%. The calculation was carried on the website of the sample size calculator: OpenEpi. The minimal sample size required for the study was 365 persons [13].

## The questionnaire

Data were collected during a face-to-face interview questionnaire. The questionnaire is a combination of items from published studies on the HPV awareness and knowledge, and from research in the literature [14, 15].

The questionnaire was divided into three main sections. The first section: was designed to collect the socio-demographic information of the participants (i.e., age, gender, marital status, employment status, geographic origin, level of study, and curriculum).

In the second section: participants were first asked to answer the following question (Q1): Do you know any sexually transmitted infections (STIs)? This was followed by a question with "Yes" or "No" response options (Q2): Have you ever heard of cervical cancer? To assess the students' awareness of HPV infection, we asked them to answer another question with "Yes" or "No" response options (Q3): Have you ever heard of the human papillomavirus (HPV)?

Only the participants that responded "Yes" to the question above (Q3) were considered aware of HPV and were asked to complete section 3 related to knowledge on HPV infection. This section included five questions with Yes/No/Don't know response options as follows: Is HPV infection symptomatic? (Q4); is HPV infection associated with any disease? (Q5); Is there any association between HPV and cervical cancer? (Q6); Do you think that HPV infection is the main cause of cervical cancer? (Q7); Do you know any prevention strategies against HPV infection? (Q8). We allocated one point for each correct answer, and zero for the incorrect or the "Don't know" response. The total score was calculated by adding up all scores for the five questions (Q4 to Q8), and possible scores range from 0 to 5. The median score of five for knowledge on HPV was used to categories high and low knowledge.

The scores higher than the median (median = 2) were categories as "High knowledge"; while "Low knowledge" referred to scores equal or lower than the median [16]. The data were archived as digital files.

## The statistical analysis

Descriptive statistics include frequencies and percentages for categorical variables, mean and standard deviation (SD) for continuous variables were calculated. Student test, Chi-square ($\chi^2$) and Fisher's exact analyses were used to identify the associated sociodemographic variables to HPV awareness, knowledge on HPV, and cervical cancer awareness. P-values $<0.05$ were considered significant.

Odds ratio (OR), p-value and 95% confidence intervals (CI) were obtained using logistic regression models. Univariate and multivariate logistic regression analyses were used to determine the significant associations between the socio-demographic variables and awareness of HPV infection, and the level of knowledge on HPV infection. Variables that exhibited statistical significance at the 0.2 level in the univariate analysis were included in the multivariate analysis using a logistic regression model adjusted for potential confounders [17]. Statistical significance was defined as p-value $< 0.05$.

The collected data were analyzed using statistical analysis software (IBM SPSS statistics version 13.0, New York, NY).

### Ethics approval and consideration

The objective of the study was fully explained to the participants before the data collection. An informed written consent was obtained from each participant. Participating in the survey was voluntary and anonymous. This research was fully approved by the Bioethics consultative commission of the Faculty of Sciences of Agadir (BECC-FSA Ref. No: FCR-CS-05/2021-0001).

## Results

### a) Sociodemographic characteristics of the participants

A total of n = 479 students were included in this study (the complete data were shown in S1 Table). The age of the population was comprised between 17 to 28 years, with a mean age at 21.82 ± 2.091. The male population was n = 243 (50.70%), and the female population was n = 236 (49.3%). The majority of the participants were full-time students n = 475 (99.20%), single n = 474 (99.00%), and came from urban areas n = 353 (73.7%). More than half of them n = 261 (54.50%) were pursuing their studies in biology curricula, and the undergraduate students accounted for n = 430 (89.80%) of all the participants (Table 1).

### b) The awareness of the population related to HPV infection and its association with cervical cancer

The results showed that the acquired immune deficiency syndrome (AIDS) is the most cited sexually transmitted infection (STI) among all the STIs mentioned by the students (Q1).

**Table 1. Characteristics of the population in relation to the awareness of HPV infection and cervical cancer.**

| Variables | Number (%) n = 479 | HPV Awareness n = 479 Have you ever heard of HPV? | | | Cervical Cancer Awareness n = 479 Have you ever heard of CC? | | |
|---|---|---|---|---|---|---|---|
| | | Yes n(%) | No n(%) | p-value | Yes n(%) | No n(%) | p-value |
| **Age (mean± SD)** | 21.82±2.09 | 22.81±2.23 | 21.71±2.04 | 0.860 | 21.86±2.14 | 21.61±1.82 | 0.176 |
| 17–21 | 213(44.5) | 12 (5.6) | 201 (94.4) | 0.004* | 167 (78.4) | 46 (21.6) | 0.103 |
| 22–28 | 266 (55.5) | 36 (13.5) | 230 (86.5) | | 224 (84.2) | 42 (15.8) | |
| **Gender** | | | | | | | |
| Men | 243 (50.7) | 23 (9.5) | 220 (90.5) | 0.681 | 181 (74.5) | 62(25.5) | < 0.001* |
| Women | 236 (49.3) | 25 (10.6) | 211 (89.4) | | 210(89.0) | 26(11.0) | |
| **Marital status** | | | | | | | |
| Single | 474 (99.00) | 47 (9.9) | 427(90.1) | 0.412 | 386(81.4) | 88(18.6) | 0.590 |
| Married | 5 (1.00) | 1 (20.0) | 4 (80.0) | | 5 (100.0) | 0 (0.0) | |
| **Employed** | | | | | | | |
| Yes | 4 (0.8) | 1 (25.0) | 3 (75.0) | 0.345 | 4 (100.0) | 0 (0.0) | 1.000 |
| No | 475 (99.20) | 47 (9.9) | 428(90.1) | | 387(81.5) | 88(18.5) | |
| **Geographic origin** | | | | | | | |
| Urban | 353 (73.70) | 37(10.5) | 316(89.5) | 0.574 | 290(82.2) | 63(17.8) | 0.620 |
| Rural | 126 (26.30) | 11 (8.7) | 115(91.3) | | 101(80.2) | 25(19.8) | |
| **Level of education** | | | | | | | |
| Undergraduate | 430 (89.80) | 31 (7.2) | 399(92.8) | <0.001* | 346(80.5) | 84(19.5) | 0.051 |
| Graduate | 49 (10.20) | 17 (34.7) | 32 (65.3) | | 45 (91.8) | 5 (8.2) | |
| **Curriculum** | | | | | | | |
| Non-biology | 261 (54.50) | 6 (2.8) | 212(97.2) | <0.001* | 163(74.8) | 55(25.2) | < 0.001* |
| Biology | 218 (45.50) | 42(16.1) | 219(83.9) | | 228(87.4) | 33(12.6) | |

n = number of participants. % percentage. HPV = Human Papillomavirus. CC = Cervical Cancer.

*Statistical significance

Among the respondents n = 459 (95.80%) mentioned AIDS in the first place, followed by syphilis n = 83 (17.30%), and hepatitis B which was mentioned by only n = 32 (6.70%) of the students. Other STIs have been mentioned such as herpes n = 4 (0.80%), *Chlamydia spp*. n = 3 (0.60%) and n = 1 *Trichomonas spp*. (0.20%). Regarding Human papillomavirus (HPV) infection, we found that only 7 students (1.50%) knew that HPV infection is an STI.

Students were also asked if they knew about cervical cancer (Q2). Surprisingly, the vast majority of them have already heard about cervical cancer (81.6%, 391 out of 479), and significant differences were reported between the group who have heard about cervical cancer and the group who haven't heard about it, in relation to: gender (p< 0.001) and curriculum (p< 0.001) (Table 1). However, only n = 48 (10.0%) replied to have already heard about HPV (Q3). These students were aware of HPV infection. Differences in HPV awareness group by age (p = 0.004), level of education (p = 0.000) and curriculum (p = 0.000) were significant (Table 1).

## C) The knowledge of the population related to HPV infection and its association with cervical cancer

The results showed that among the students who previously have heard about HPV infection, n = 29 (60.4%) knew that HPV infection is asymptomatic (Q4). Only 22 out of 48 students (45.8%) knew that HPV infection can be associated with other diseases (Q5). The link between HPV and cervical cancer was confirmed by n = 23 (47.9%) of the participants (Q6). More than half n = 26 (54.2%) responded that infection with HPV is the leading cause for the development of cervical cancer (Q7). Finally, the majority of the students n = 33 (68.8%) were unable to suggest any prevention strategy against the HPV infection (Q8). Regarding the knowledge about HPV infection, the participants could be divided into two groups: Low knowledge group n = 29 (60.4%) and high knowledge group n = 19 (49.6%) (Table 2).

## D) Factors affecting the awareness and the knowledge associated to HPV infection

To investigate the association between the awareness about HPV infection and some key features of the population, we performed univariate logistic regression statistical analyses (S2 Table).

This analysis showed that HPV awareness correlated positively with the age of the participants, their curriculum, and their level of study.

The awareness about HPV infection was higher in the population of the students aged between 22 and 28 years, compared to those aged between 17 and 21 years old (13.5% *vs* 5.6%) (OR = 2.62; 95% CI = [1.32–5.17]; p = 0.005). In addition, graduate students were more likely to be aware of HPV infection compared to undergraduate students (OR 6.83; 95% CI = [3.42–13.66] p <0.001). Students with a biology curriculum knew more about HPV compared to non-biology curriculum students (16.1% *vs* 2.8%) (OR = 6.77; 95% CI = [2.82–16.27]; p <0.001). To confirm these results, we entered the most significant variables into a multivariate model. This model showed that the high level of education (OR = 4.04; 95% CI = [1.92–8.52]; *p* = 0.001), and the nature of the curriculum (OR = 4.92; 95% CI = [2.12–12.73]; p = 0.001) remained significantly associated factors with HPV awareness (Table 3).

The univariate logistic regression analysis revealed that knowledge about HPV was associated significantly with the gender and the level of study (S2 Table). Females were more likely to have higher knowledge about HPV compared to males (56.0% *vs* 21.7%) (OR = 4.58; 95% CI = [1.29–16.26]; *p* = 0.019). Education level had a positive effect on the knowledge related to HPV, graduate students were more likely to have better knowledge of HPV compared to

**Table 2. Characteristics of the population in correlation with the knowledge related to HPV and cervical cancer.**

| Variables | Number (%) n = 479 | Knowledge on HPV n = 48 | |
|---|---|---|---|
| | | Low n(%) n = 29 | High n(%) n = 19 |
| **Age** | | | |
| Mean ± SD | 21.82 ± 2.091 | 22.69±2.34 | 23±2.10 |
| 17–21 | 213 (44.50) | 7 (58.3) | 5 (41.7) |
| 22–28 | 266 (55.50) | 22 (61.1) | 14 (38.9) |
| **Gender** | | | |
| Men | 243 (50.70) | 18 (78.3) | 5 (21.7) |
| Women | 236 (49.30) | 11 (44.0) | 14 (56.0) |
| **Marital status** | | | |
| Single | 474 (99.00) | 29 (61.7) | 18 (38.3) |
| Married | 5 (1.00) | 0 (0.0) | 1 (100.0) |
| **Employed** | | | |
| Yes | 4 (0.8) | 0 (0.0) | 1 (100.0) |
| No | 475 (99.20) | 29 (61.7) | 18 (38.3) |
| **Geographic origin** | | | |
| Urban | 353 (73.70) | 21 (56.8) | 16 (43.2) |
| Rural | 126 (26.30) | 8 (72.7) | 3 (27.3) |
| **Level of education** | | | |
| Undergraduate | 430 (89.80) | 22 (71.0) | 9 (29.0) |
| Graduate | 49 (10.20) | 7 (41.2) | 10 (58.8) |
| **Curriculum** | | | |
| Non-biology | 261 (54.50) | 23 (54.8) | 19 (45.2) |
| Biology | 218 (45.50) | 6 (100.0) | 0 (0.0) |

n = number of participants. % percentage. HPV = Human Papillomavirus. CC = Cervical Cancer. *Statistical significance.

undergraduate students (OR = 3.49; 95% CI = [1.01–12.05]; p = 0.048). The multivariate analysis showed that only gender remained significantly associated factor with HPV knowledge (OR = 3.76; 95% CI = [1.01–13.92]; p = 0.047) (Table 3).

## Discussion

Very limited reports investigated the awareness of STIs, particularly the HPV infection and its association with cervical cancer among higher education students in Morocco, despite this part of the population was the most concerned by such issues.

As a first approach to assess the awareness of HPV infection, the students were asked to indicate which STIs they already knew. This question was mainly asked to find out if the students can identify HPV infection among STIs. We showed that the majority of students have little knowledge concerning STIs. In addition, very few of them knew that HPV infection is STI (1.50%). However, a high proportion of them (95.8%) were aware that AIDS is STI. AIDS was the best known STI, which might be the result of the massive educative programs through the media, schools, and health professionals which enhanced awareness of the general population about it. Similarly, it was reported in Egypt that 95.9% of the students knew about AIDS, but they showed a poor knowledge about sexual health issues [18]. By contrast, students in the UK had a higher level of knowledge about STIs [19]. This might be explained by the effect of cultural differences and religious sensitivities between countries, which limited considerably sexual education and awareness among students. Among the students, 81.6% were aware of

**Table 3. Multivariate logistic regression analysis to examine the factors associated with HPV awareness and knowledge.**

| Variables | HPV Awareness | | HPV Knowledge | |
|---|---|---|---|---|
| | Multivariate analysis | | Multivariate analysis | |
| | aOR (CI 95%) | p-value | aOR (CI 95%) | p-value |
| **Age categories** | | | | |
| 17–21 | 1 | 0.096 | - | - |
| 22–28 | 1.85 (0.89–3.83) | | | |
| **Gender** | | | | |
| Men | - | - | 1 | 0.047* |
| Women | | | 3.76 (1.01–13.92) | |
| **Level of education** | | | | |
| Undergraduate | 1 | < 0.001* | 1 | 0.145 |
| Graduate | 4.04 (1.92–8.52) | | 2.65 (0.71–9.81) | |
| **Curriculum** | | | | |
| Non-biology | 1 | < 0.001* | - | - |
| Biology | 5.20 (2.12–12.73) | | | |

%percentage. HPV = Human Papillomavirus, CI = Confidence interval. aOR = Adjusted Odds ratio.

1: Reference category.

*Statistical significance

cervical cancer. This awareness was lower than the 94.2% obtained in the study conducted in the Netherlands, however, higher than the 56.4% reported among students in Nigeria and 33.3% in South Africa (33%) [20–22]. The existence of effective educational programs to raise awareness of cervical cancer in the European countries might explain the high level of awareness compared to the African countries.

In this study, only 10.0% of the students reported having heard about HPV and showed a poor level of HPV knowledge. This result was lower than a study conducted among Moroccan students (33.7%), but they also showed a poor level of HPV knowledge [15]. This disparity in HPV awareness in the same country can be explained by the fact that students in southern Morocco live in areas with fewer healthcare facilities. Such low HPV awareness has also been reported by several studies, among university students in Nigeria (17.7%), Portugal (55.4%), Malaysia (21.7%), in The Netherlands (17.7%), among female medical students in Ethiopia (50.6%), and they all showed a low level of HPV knowledge [20, 21, 23–25]. Contrary to this finding, a study carried out in the USA reported that (78.0%) had heard of HPV and showed a relatively high HPV knowledge [26]. Besides that, our finding associated more HPV awareness with being a biology student and having high educational attainment. The study conducted in Morocco, the authors agreed that education level had a positive effect on HPV awareness [15]. A similar finding was reported by a study in India where biology students have shown more awareness towards HPV when compared to non-biology students [14]. A significant association was found between being a health science student and having heard about HPV in the study conducted in Portugal [23]. This had been also agreed by a study in the Netherlands where medical students were more aware of HPV than non-medical students [20]. These results suggest a misunderstanding or lack of sexual health background among non-biology students which explains why they showed low HPV awareness compared to biology students. It also seems plausible that students with high educational levels have more chances to hear about HPV than those with lower educational levels. We might find a positive correlation between the age and their level of education. However, it remains questionable at what extent

age can be regarded as an independent risk factor functioning as confounder in the awareness-level of education relationship.

Our finding also revealed that women showed better knowledge about HPV infection than men did, which is consistent with the finding reported by researchers in Portugal [23]. The poor HPV knowledge among men highlights that they are less knowledgeable about sexual health issues than women. Indeed, it is very important to intensify HPV knowledge among all the students, especially men because they participate in the transmission and the acquisition of HPV infection.

This present study highlights a poor awareness and knowledge towards STIs in particular HPV infections among university students. Those students who were supposed to be the most knowledgeable showed a big gap in the understanding of their sexual health. Our findings suggest the urgent need for well-designed educational interventions and university campaigns to improve HPV, STIs, and cervical cancer awareness. Targeting higher education students as future educators, parents, and community members will guarantee the next generations with more awareness towards HPV and its negative outcomes.

## Conclusion

In the current study, we showed that, the majority of the students 391 out of 479 have heard of cervical cancer, but only 23 out of 479 students were aware that HPV infection is associated with the development of cervical cancer.

Indeed, we found that the most significant factors associated with better awareness about HPV were respectively: (i) the high level of education; (ii) the attendance to curricula related to biological science, while the only associated factor with high knowledge on HPV was the female gender.

Accordingly, we believe that the early incorporation of sexually communicable diseases teaching programs in higher education in Morocco will improve the awareness of young educated people, and reduce the burden of HPV associated diseases among this population at risk.

## Supporting information

**S1 Table. Study data n = 479 participants.**
(XLSX)

**S2 Table. Logistic regression analysis: Detailed univariate and multivariate analysis of the data.**
(DOCX)

## Acknowledgments

We would like to thank all the students that kindly agreed to participate in this study.

## Author Contributions

**Conceptualization:** Nezha El Mansouri, Mohamed Nejmeddine.

**Data curation:** Nezha El Mansouri.

**Formal analysis:** Nezha El Mansouri, Laila Ferrera, Ahmed Kharbach.

**Investigation:** Nezha El Mansouri, Hanane Rogua, Sofiane Ait Wahmane.

**Methodology:** Nezha El Mansouri, Ahmed Kharbach, Abderrahmane Achbani.

**Project administration:** Mohamed Nejmeddine.

**Resources:** Nezha El Mansouri.

**Software:** Nezha El Mansouri, Ahmed Kharbach.

**Supervision:** Mohamed Nejmeddine.

**Validation:** Nezha El Mansouri, Ahmed Kharbach, Abderrahmane Achbani.

**Visualization:** Nezha El Mansouri, Laila Ferrera.

**Writing – original draft:** Nezha El Mansouri.

**Writing – review & editing:** Nezha El Mansouri, Laila Ferrera, Ahmed Kharbach, Abderrahmane Achbani, Farid Kassidi, Hanane Rogua, Sofiane Ait Wahmane, Ahmed Belmouden, Said Chouham, Mohamed Nejmeddine.

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
