## [Decision Letter · Decision Letter 0]

25 Jan 2022

PONE-D-21-38027Knowledge and Awareness toward Human Papillomavirus and Cervical Cancer: A Cross-sectional Study Among University Students in Morocco.PLOS ONE

Dear Dr. EL MANSOURI,

Thank you for submitting your manuscript to PLOS ONE. After careful consideration, we feel that it has merit but does not fully meet PLOS ONE’s publication criteria as it currently stands. Therefore, we invite you to submit a revised version of the manuscript that addresses the points raised during the review process. Please submit your revised manuscript by Mar 11 2022 11:59PM. If you will need more time than this to complete your revisions, please reply to this message or contact the journal office at plosone@plos.org. Please include the following items when submitting your revised manuscript:A rebuttal letter that responds to each point raised by the academic editor and reviewer(s). You should upload this letter as a separate file labeled 'Response to Reviewers'.A marked-up copy of your manuscript that highlights changes made to the original version. You should upload this as a separate file labeled 'Revised Manuscript with Track Changes'.An unmarked version of your revised paper without tracked changes. You should upload this as a separate file labeled 'Manuscript'.

We look forward to receiving your revised manuscript.

Kind regards,

Muhammad Tarek Abdel Ghafar, M.D

Academic Editor

PLOS ONE

Journal Requirements:

Reviewers' comments:

Reviewer's Responses to Questions

**Comments to the Author**

1. Is the manuscript technically sound, and do the data support the conclusions?

Reviewer #1: Yes

Reviewer #2: Yes

Reviewer #3: Yes

Reviewer #4: No

2. Has the statistical analysis been performed appropriately and rigorously? 

Reviewer #1: Yes

Reviewer #2: Yes

Reviewer #3: Yes

Reviewer #4: Yes

3. Have the authors made all data underlying the findings in their manuscript fully available?

Reviewer #1: Yes

Reviewer #2: No

Reviewer #3: Yes

Reviewer #4: Yes

4. Is the manuscript presented in an intelligible fashion and written in standard English?

Reviewer #1: Yes

Reviewer #2: Yes

Reviewer #3: Yes

Reviewer #4: No

5. Review Comments to the Author

Reviewer #1: The manuscript is well written and has potential.

However, a few typographical and technical errors have been noted in the attached document.

Some highlights to address:

1. Please clearly state your sample size

2. Maintain one style of citation

Good job

Reviewer #2: The manuscript provides a logical flow from the introduction to the study objectives and conclusion.

The manuscript is well presented and written in standard comprehensible English. However, it can still benefit from structural rearrangement of some of the sentences to make readability easier, especially the result and discussion section.

It was mentioned that the study data is available without restriction but information on where and how it could be accessed was not provided.

The section on material and methods: More clarity should be provided abut the mode of questionnaire administration. How was this done? (self-administration, facilitated or online assessment). This need to be clear. The section on the questionnaire could probably focused more on the description of the questionnaire and approach to the response which are more useful for replicability rather than on specific content of the questionaire

Reviewer #3: Thanks dear authors. The topic and findings are revelant. Kindly find my comments for improvment and question that need clarification.

1. Title:

Knowledge and Awareness toward Human Papillomavirus and Cervical Cancer(CC): A Cross-sectional Study Among University Students in Morocco. This does not suggest that you will find the factors associated with awarness and knowelege about HPV and CC.

2. Introduction:

To what extent are university students at risk for HPV related infections such as cervical cancer? This information can justfy your research if included in the introduction.

3. Methods section:

how do calculate the sample size of 479 students and allocate in to faculities and lebow? Describe the smaple size determination and prcudure.

The way you describe measurement of knowledge in your research is intersting. However, the cut of poit 3 correct answers needs a reference or justification?

4. Results:

The presentation of your result shall be according to your objectives. for example

a) describe sociademographic and reproductive charactrestics of your participants

b) their awarness about HPV and CC

C) Their knowledge about HPV andCC

D) Factors affecting their knowledge

So tabel 1 need to be split

Why class year is not recoreded?as long as awarness of the first year and graduating class student won't be the same

On table2: the AOR foe age and level of study are mispolaced. correct them. again the insteady of writting ref, use 1

In table2: the uni and multivariate analusis result should be presented separately if you must report. For me Univariate is not needed. The p- value and confidence interval should be reported in the multivariate analysis.You need to presnt also ont only about HPV but also CC. In general table 2 can not be interpretued correctly, there are missing elemests. i.e the CI sould be consistent with the response of the outcome variable. This needs revision

5 Discussion: is good

6. references

Replace references older than 2017 for article or 2002 for books with recent onces

Reviewer #4: Critical review of the manuscript titled

”Knowledge and Awareness toward Human Papillomavirus and Cervical Cancer: A Cross-sectional Study Among University Students in Morocco”

submitted to PLOS ONE

The authors report the results of a questionnaire survey on awareness and knowledge of HPV infection, cervical cancer and their relationship, conducted among university students in Agadir, Morocco. The topic is an important public health issue but the coverage of the study is very limited, the study population is not representative even for the students at the university and the development of the questionnaire is unclear.

Major comments

1) “The participants were selected using a simple random selection process. However, the population of students targeted was mainly those attending courses in the Faculty of Sciences.” (page 4, lines 20-21). The two sentences contradict each other. The source population is unclear; therefore, the study population seems to be not representative for any population group.

2) “This was followed by an open-ended question with “Yes” or “No” response options…” (page 5, lines 6-7). A question with Yes/No options is not an open-ended question.

3) The development of the survey tool is not explained. Is it a standardized questionnaire? Was it pilot tested? Were questions from other surveys adopted? How was the cut-off level of knowledge determined?

4) Why was age dichotomized and not handled as a continuous variable? This approach introduced loss of information.

5) How could it happen that fewer participants knew the association between HPV and cervical cancer than those who knew that HPV is its leading cause? It seems a contradiction.

6) Age essentially correlates with the level of education; therefore, it is questionable whether age can be regarded as an independent risk factor functioning as confounder in the awareness-level of education relationship.

Minor comments

1) The aim of the study is not clearly phrased in the Abstract.

2) “Logistic regression analyses were to determine...” (page 2, line 8). Insert “used”.

3) “…higher level of study…” (page 2, line 14 and at several other places). I suggest using “level of education” throughout.

4) “University student” (Keywords) should not be capitalized.

5) “Bivalent (HPV-16 and HPV-18), and quadrivalent (HPV-16, HPV-18, HPV-6, and HPV-11), and nonavalent vaccine targets the same HPV types as the quadrivalent vaccine as well as types (31, 33, 45, 52, and 58) [8,9].” (page 3, lines 12-14). The sentence is incorrectly phrased.

6) “…for categorical variables. Mean and…” (page 5, line 21) should be “…for categorical variables, mean and…”.

7) “adjusted by potential confounders” (page 6, line 4) should be “adjusted for potential confounders”.

8) “[Table 1]” (page 7, line 8) should be “(Table 1).

9) Asterisk should rather be used to highlight significance in Table 1.

10) There is no point to report p value if CI95% is reported in Table 2.

11) “…(50.6%) among female medical students in Ethiopia [29],…” (page 12, line 5) should be “…among female medical students in Ethiopia (50.6%) [29],…”.

12) “…women showed good knowledge…” (page 12, line 18) should be “…women showed better knowledge…”.

Summary

The manuscript discusses an important public health topic but the study setting is very limited, not representative, and the development of the survey instrument is unclear. Under these circumstances, the results cannot lead to solid scientific conclusions.

6. PLOS authors have the option to publish the peer review history of their article (what does this mean?). If published, this will include your full peer review and any attached files.

Reviewer #1: No

Reviewer #2: No

Reviewer #3: No

Reviewer #4: No

---

## [Author Response · Author response to Decision Letter 0]

25 Feb 2022

Dear Muhammad Tarek Abdel Ghafar,

I would like to thank you and the reviewers for the useful comments on our article. I now have edited the manuscript to address the concerns raised by the reviewers. Accordingly, please find below the response, point by point.

I hope that the manuscript, in its actual form, meets the standards for publication in PLOS ONE. 

I am looking forward to hearing from you soon,

Sincerely yours,

EL MANSOURI Nezha.

 

Academic editor:

Point 1: 

Corrected. We hopefully have no divergences from the style requirements now.

Point 2:

Please provide additional details regarding participant consent. In the ethics statement in the Methods and online submission information, please ensure that you have specified what type you obtained (for instance, written or verbal, and if verbal, how it was documented and witnessed).

We amended the ethics statements to emphasize that all participants gave written consent, line 136-137 (Simple Markup track changes).

An informed written consent was obtained from each participant when they agreed to be included in this study. The ethics statement was also amended in the submission form.

Point 3:

We note that you have stated that you will provide repository information for your data at acceptance. Should your manuscript be accepted for publication, we will hold it until you provide the relevant accession numbers or DOIs necessary to access your data. If you wish to make changes to your Data Availability statement, please describe these changes in your cover letter and we will update your Data Availability statement to reflect the information you provide.

Thank you for the remark. 

I would like to let you know that the box referring to “the additional data availability information” was ticked by mistake during the submission online. We removed this tick mark. Instead, all relevant data are now available in the manuscript and in the supporting information files.

Reviewer #1

The manuscript is well written and has potential. However, a few typographical and technical errors have been noted in the attached document.

All highlighted typographical and technical errors have been corrected. We thank the reviewer for helping us.

Some highlights to address:

Point 1: 

Please clearly state your sample size

Amended, we added the following sentence in line 92 (Simple markup track changes): A total of n= 479 participants were selected using a simple random selection process. 

Point 2:

Maintain one style of citation

Amended.

Reviewer #2

Point 1: 

The manuscript is well presented and written in standard comprehensible English. However, it can still benefit from structural rearrangement of some of the sentences to make readability easier, especially the result and discussion section.

Amended.

Point 2:

It was mentioned that the study data is available without restriction but information on where and how it could be accessed was not provided.

We have sent the study data (Excel file) as a supporting information file.

The section on material and methods: 

Point 3:

More clarity should be provided about the mode of questionnaire administration. How was this done? (self-administration, facilitated or online assessment). This need to be clear. The section on the questionnaire could probably focused more on the description of the questionnaire and approach to the response which are more useful for replicability rather than on specific content of the questionnaire.

The administration of the questionnaire was a face-to-face interview questionnaire done by our team members, who also helped in clarifying questions for the students. We added a sentence in the manuscript that emphasize the type of questionnaire and how it was implemented on line 99 (simple markup track changes). 

Reviewer #3

Point 1:

Title: Knowledge and Awareness toward Human Papillomavirus and Cervical Cancer (CC): A Cross-sectional Study Among University Students in Morocco. This does not suggest that you will find the factors associated with awareness and knowledge about HPV and CC.

Thank you for your comment. We have changed the title to reflect more the study objectives. 

Corrected title: “Human Papillomavirus knowledge, awareness and associated factors: A cross-sectional study among university students in Morocco.”

Point 2:

Introduction: To what extent are university students at risk for HPV related infections such as cervical cancer? This information can justify your research if included in the introduction.

Cervical cancer is a consequence of the persistence of HPV infection which is the most common sexually transmitted infection worldwide. The highest rates of HPV infection are found among young and sexually active men and women. It was also shown that HPV infection prevalence decreased with increasing age from a peak prevalence in younger women (≤25 years of age) (Smith JS et al., Age-Specific Prevalence of Infection with Human Papillomavirus in Females: A Global Review. J Adolesc Heal. 2008 43:5–25). Students at the university are young adults and at the age of sexual activity, and the probability of contracting HPV infection during this period is high. We have added this information in the introduction on lines 52-53 and 87 (simple markup track changes).

Point 3:

Methods section: how do calculate the sample size of 479 students and allocate in to faculities and lebow? Describe the sample size determination and prcudure. 

We have added information on the manuscript to clarify the sample size calculation, lines 93-97 (simple markup track changes). As follow:

The size of the sample was calculated based on a 5.0% error range, a 95% confidence interval (CI) for a total population of 7000 students in the faculty of sciences, university Ibn Zohr. With an anticipated proportion of knowledge and awareness on HPV deficiency of 50%. The sample size was calculated using the website OpenEpi https://www.openepi.com/Menu/OE_Menu.htm. The minimal sample size required for the study was 365 persons.

The way you describe measurement of knowledge in your research is interesting. However, the cut of point 3 correct answers needs a reference or justification?

We have added information in the manuscript to clarify the cut of point 3, on lines 115-121 (simple markup track changes), as follow: 

We allocated one point for each correct answer, and zero for the incorrect or the “Don’t know” responses. The total score was calculated by adding up all scores for the five questions (Q4 to Q8), and possible scores range from 0 to 5. The scores higher than the median (median=2) (Higher or equal than 3 correct answers) were designated as categories of “High knowledge”. By contrast, “Low knowledge” referred to scores equal or lower than the median or (Lower than 3 correct answers) (Tusimin M et al. Sociodemographic determinants of knowledge and attitude in the primary prevention of cervical cancer among University Tunku Abdul Rahman (UTAR) students in Malaysia: Preliminary study of HPV vaccination. BMC Public Health. 2019;19(1):1–6).

Point 4:

Results: The presentation of your result shall be according to your objectives. For example: 

a) describe sociodemographic and reproductive characteristics of your participants

b) their awareness about HPV and CC

C) Their knowledge about HPV and CC

D) Factors affecting their knowledge

So table 1 need to be split

We thank the reviewer for the constructive feedback. All the suggested changes in the results section structure and table 1 have been made. Table 1 is split into Tables 1 and 2, on pages 7 and 8 respectively.

However, regarding point (c), our work was mainly focused on what causes CC, instead of the knowledge toward CC. 

Why class year is not recorded? as long as awareness of the first year and graduating class student won't be the same

We did record class year for each student during the data collection as follow: 1st year, 2nd year, 3rd year and masters, and then we considered (1st year, 2nd year, 3rdyear = undergraduate and masters= graduate). Indeed, logistic regression is the type of analysis to use when we work with binary data. Therefore, we categorized all the independent variables in our study into two groups. We omitted this details in the manuscript in order to keep the independent variable (Level of education) dichotomous.

On table2: the OR for age and level of study are misplaced. correct them. again the instead of writing ref, use 1

Corrected.

In table2: the uni and multivariate analysis result should be presented separately if you must report. For me Univariate is not needed. The p- value and confidence interval should be reported in the multivariate analysis. You need to present also ont only about HPV but also CC. In general table 2 cannot be interpreted correctly, there are missing elements. i.e the CI should be consistent with the response of the outcome variable. This needs revision.

As required above, we have modified the table accordingly (Table 2 is now Table 3 on page 10). 

References: Replace references older than 2017 for article or 2002 for books with recent ones

As required, the references were updated in the introduction section. 

Reviewer #4

Major comments

Point 1: 

“The participants were selected using a simple random selection process. However, the population of students targeted was mainly those attending courses in the Faculty of Sciences.” (page 4, lines 20-21). The two sentences contradict each other. The source population is unclear; therefore, the study population seems to be not representative for any population group. 

Thank you for your comment. We have corrected this in the manuscript on line 90-92 as follow:

- This cross-sectional study was conducted on the site of the faculty of sciences, Ibn Zohr University (IZU), in Agadir, Morocco, where the majority of the students were attending their curricula. A total of n= 479 participants were included using a simple random selection process. 

- The choice of the population: In fact, university students are young adults and at the age of sexual activity, making them at high risk of contracting HPV infection.

Point 2:

 “This was followed by an open-ended question with “Yes” or “No” response options…” (page 5, lines 6-7). A question with Yes/No options is not an open-ended question.

Corrected in the manuscript on lines 106, 108 and 112 (Simple markup track changes).

Point 3:

The development of the survey tool is not explained. Is it a standardized questionnaire? Was it pilot tested? Were questions from other surveys adopted? 

The questionnaire is a combination of items adapted from previously published studies below:

- Zouheir Y et al., Knowledge of Human Papillomavirus and Acceptability to Vaccinate in Adolescents and Young Adults of the Moroccan Population. 

J Pediatr Adolesc Gynecol. 2016;29(3):292–8; 

- Rashid S et al., Knowledge, awareness and attitude on HPV, HPV vaccine and cervical cancer among the college students in India. 

PLoS One. 2016;11(11):1–11.) 

We added this modifications in the manuscript on lines: 99-101.

How was the cut-off level of knowledge determined? 

We have added information in the manuscript to clarify the cut of point 3, on lines 115-121, (simple markup track changes), as follow: 

We allocated one point for each correct answer, and zero for the incorrect or the “Don’t know” responses. The total score was calculated by adding up all scores for the five questions (Q4 to Q8). The possible scores range is from 0 to 5. The scores higher than the median (median=2; Higher or equal than 3 correct answers) were considered as categories of “High knowledge”. By contrast, “Low knowledge” referred to scores equal or lower than the median or (Lower than 3 correct answers) (Tusimin M et al. Sociodemographic determinants of knowledge and attitude in the primary prevention of cervical cancer among University Tunku Abdul Rahman (UTAR) students in Malaysia: Preliminary study of HPV vaccination. BMC Public Health. 2019;19(1):1–6)).

Point 4:

Why was age dichotomized and not handled as a continuous variable? This approach introduced loss of information.

Corrected. The independent t-test was performed when we added age as a continuous variable (Table 1 and Table 2.). However, in the logistic regression analysis age was dichotomised based on the mean to facilitate the analysis (Table 3). 

Point 5:

 How could it happen that fewer participants knew the association between HPV and cervical cancer than those who knew that HPV is its leading cause? It seems a contradiction. 

The present study reveals that 60.4% of the students who have heard about HPV infection, did show a low knowledge about the virus itself. Those students, through their responses, showed that there is a gap in their understanding of this infection and of its consequences (i.e. association between HPV and CC). In addition, due to the nature of the answers proposed (i.e. Yes/ No/ Don’t know) we might have overestimated the level of knowledge of our surveyed population. This may constitute one of the limitations of this study. Indeed, open questions are needed to avoid influencing the answers given by the participants.

Point 6:

Age essentially correlates with the level of education; therefore, it is questionable whether age can be regarded as an independent risk factor functioning as confounder in the awareness-level of education relationship.

We agree that level of education correlate with age; this has been emphasized in the discussion section, on lines 247-250 (Simple markup track changes). 

Minor comments

We thank the reviewer for spotting these errors and for their constructive feedback. All the suggested minor comments have been corrected and highlighted in the manuscript (red font).

---

## [Decision Letter · Decision Letter 1]

14 Mar 2022

PONE-D-21-38027R1Human Papillomavirus knowledge, awareness and associated factors: A cross-sectional study among university students in MoroccoPLOS ONE

Dear Dr. EL MANSOURI,

Thank you for submitting your manuscript to PLOS ONE. After careful consideration, we feel that it has merit but does not fully meet PLOS ONE’s publication criteria as it currently stands. Therefore, we invite you to submit a revised version of the manuscript that addresses the points raised during the review process.

We look forward to receiving your revised manuscript.

Kind regards,

Muhammad Tarek Abdel Ghafar, M.D

Academic Editor

PLOS ONE

Journal Requirements:

Reviewers' comments:

Reviewer's Responses to Questions

**Comments to the Author**

1. If the authors have adequately addressed your comments raised in a previous round of review and you feel that this manuscript is now acceptable for publication, you may indicate that here to bypass the “Comments to the Author” section, enter your conflict of interest statement in the “Confidential to Editor” section, and submit your "Accept" recommendation.

Reviewer #2: All comments have been addressed

Reviewer #3: All comments have been addressed

2. Is the manuscript technically sound, and do the data support the conclusions?

Reviewer #2: Yes

Reviewer #3: Yes

3. Has the statistical analysis been performed appropriately and rigorously? 

Reviewer #2: Yes

Reviewer #3: Yes

4. Have the authors made all data underlying the findings in their manuscript fully available?

Reviewer #2: Yes

Reviewer #3: Yes

5. Is the manuscript presented in an intelligible fashion and written in standard English?

Reviewer #2: Yes

Reviewer #3: Yes

6. Review Comments to the Author

Reviewer #2: Most of the comments have been addressed.

Need for a minor correction at Line 105. A reference to a "Yes" or "No" question is not open ended.

Reviewer #3: There are still some comments

1. knowledge comes after awarness. analysining the associated factors for awarness and knowledge is just repeation.

2. The tabular presention needs correction

The logical order of presention should be according to your objectives. first description of study participants, then knowedge, then the factors. hte others are not relevant. try to describe what is the levele of knowledge about HPV/CC and the associated factors. The Chi Square is not important. you already catagorise your dependent variable knowledge as high and low. So, you can use logistic regression.

7. PLOS authors have the option to publish the peer review history of their article (what does this mean?). If published, this will include your full peer review and any attached files.

Reviewer #2: No

Reviewer #3: No

---

## [Author Response · Author response to Decision Letter 1]

27 Apr 2022

Dear Muhammad Tarek Abdel Ghafar,

I would like to thank you and the reviewers for the useful comments on our manuscript. I now have edited the manuscript to address the concerns raised by the reviewers. Accordingly, please find below the response, point by point.

I hope that the manuscript in its actual form meets the standards for publication in PLOS ONE. 

I am looking forward to hearing from you soon,

Sincerely yours,

EL MANSOURI Nezha.

 

Academic editor:

Please review your reference list to ensure that it is complete and correct.

Amended. Thank you.

 we have completed the missing information for each reference. We replaced the reference number [13] on lines 307-309 ( (simple markup track changes) : OpenEpi:Sample Size for X-Sectional,Cohort,and Clinical Trials [Internet]. [cited 2022 Feb 21]. Available from: http://www.openepi.com/SampleSize/SSCohort.htm, with Sullivan KM, Dean A, Minn MS. OpenEpi: A web-based epidemiologic and statistical calculator for public health. Public Health Reports. 2009; 124(3): 471–474. doi: 10.1177/003335490912400320

Reviewer #2

 Need for a minor correction at Line 105. A reference to a "Yes" or "No" question is not open ended.

Amended. Thank you.

Reviewer #3

Point 1

Knowledge comes after awareness. 

Amended

 Corrected title: “Awareness and knowledge associated to Human Papillomavirus infection among university students in Morocco: A cross-sectional study”.

Point 3

Analysing the associated factors for awareness and knowledge is just repetition.

In the present study awareness and knowledge related to HPV infection were investigated separately, as well as the associated factors. Awareness on HPV was investigated using a Yes or No question “Have you ever heard of Human Papillomavirus (HPV)? This question was asked to the whole population included in this study. By contrast, the knowledge related to HPV infection was investigated exclusively among students that responded “Yes” to the previous question (i.e. Have you ever heard of Human Papillomavirus?)

Point 3

The tabular presentation needs correction.

The logical order of presentation should be according to your objectives.

First description of study participants, then knowledge, then the factors. The others are not relevant. Try to describe what is the level of knowledge about HPV/CC and the associated factors. 

We investigated the level of knowledge related to HPV infection and the factors that might be associated. However, the adopted questionnaire did not investigate specifically the level of knowledge related to cervical cancer. Indeed, our investigation focused on awareness and knowledge related only to HPV infection and the factors that might be associated. 

The Chi Square is not important. you already categorise your dependent variable knowledge as high and low. So, you can use logistic regression.

Amended on lines 212-214 (simple markup track changes), and in table 2.

 Thank you.

---

## [Editor Report · Decision Letter 2]

12 Jun 2022

PONE-D-21-38027R2Awareness and knowledge associated to Human Papillomavirus infection among university students in Morocco: A cross-sectional studyPLOS ONE

Dear Dr. EL MANSOURI,

Thank you for submitting your manuscript to PLOS ONE. After careful consideration, we feel that it has merit but does not fully meet PLOS ONE’s publication criteria as it currently stands. Therefore, we invite you to submit a revised version of the manuscript that addresses the points raised during the review process.

We look forward to receiving your revised manuscript.

Kind regards,

Muhammad Tarek Abdel Ghafar, M.D

Academic Editor

PLOS ONE

Journal Requirements:

Additional Editor Comments:

Please cite the supplementary tables within the text.
---

## [Author Response · Author response to Decision Letter 2]

23 Jun 2022

Academic editor:

Please cite the supplementary tables within the text.

Amended, we cited the supplementary tables on lines 142-143, 187 and 200 (simple markup track changes).

---

## [Editor Report · Decision Letter 3]

27 Jun 2022

Awareness and knowledge associated to Human Papillomavirus infection among university students in Morocco: A cross-sectional study

PONE-D-21-38027R3

Dear Dr. EL MANSOURI,

We’re pleased to inform you that your manuscript has been judged scientifically suitable for publication and will be formally accepted for publication once it meets all outstanding technical requirements.

Kind regards,

Muhammad Tarek Abdel Ghafar, M.D

Academic Editor

PLOS ONE
---

## [Editor Report · Acceptance letter]

30 Jun 2022

PONE-D-21-38027R3 

Awareness and knowledge associated to Human Papillomavirus infection among university students in Morocco: A cross-sectional study  

Dear Dr. EL MANSOURI:

I'm pleased to inform you that your manuscript has been deemed suitable for publication in PLOS ONE. Congratulations! Your manuscript is now with our production department. 

Kind regards, 

on behalf of

Prof Muhammad Tarek Abdel Ghafar 

Academic Editor

PLOS ONE